# Impact of Leptin on the Expression Profile of Macrophages during Mechanical Strain In Vitro

**DOI:** 10.3390/ijms231810727

**Published:** 2022-09-14

**Authors:** Eva Paddenberg, Hannah Osterloh, Jonathan Jantsch, Andressa Nogueira, Peter Proff, Christian Kirschneck, Agnes Schröder

**Affiliations:** 1Department of Orthodontics, University Hospital Regensburg, 93053 Regensburg, Germany; 2Department of Microbiology and Hygiene, University Hospital Regensburg, 93053 Regensburg, Germany; 3Department of Periodontology and Operative Dentistry, University Medical Center of the Johannes Gutenberg University, 55131 Mainz, Germany

**Keywords:** macrophages, inflammation, orthodontic tooth movement, leptin, mechanical strain

## Abstract

Childhood obesity is a growing problem in industrial societies and associated with increased leptin levels in serum and salvia. Orthodontic treatment provokes pressure and tension zones within the periodontal ligament, where, in addition to fibroblasts, macrophages are exposed to these mechanical loadings. Given the increasing number of orthodontic patients with these conditions, insights into the effects of elevated leptin levels on the expression profile of macrophages during mechanical strain are of clinical interest. Therefore, the aim of this in vitro study was to assess the influence of leptin on the expression profile of macrophages during simulated orthodontic treatment. RAW264.7 macrophages were incubated with leptin and lipopolysaccharides (LPS) from *Porphyromonas gingivalis* (*P. gingivalis*) or with leptin and different types of mechanical strain (tensile, compressive strain). Expression of inflammatory mediators including tumor necrosis factor (TNF), Interleukin-1-B (IL1B), IL6, and prostaglandin endoperoxide synthase (PTGS2) was assessed by RT-qPCR, ELISAs, and immunoblot. Without additional mechanical loading, leptin increased *Tnf*, *Il1b*, *Il6*, and *Ptgs2* mRNA in RAW264.7 macrophages by itself and after stimulation with LPS. However, in combination with tensile or compressive strain, leptin reduced the expression and secretion of these inflammatory factors. By itself and in combination with LPS from *P. gingivalis*, leptin has a pro-inflammatory effect. Both tensile and compressive strain lead to increased expression of inflammatory genes. In contrast to its effect under control conditions or after LPS treatment, leptin showed an anti-inflammatory phenotype after mechanical stress.

## 1. Introduction

Childhood obesity is a growing problem in industrial societies [1,2]. In obesity, there is an increased release of endocrine proteins and metabolic dysregulation with a shift in balance in favor of anti-inflammatory adipokines. This suggests a low-grade chronic inflammation in the body. Due to an altered immune response, obesity is considered a major risk factor for systemic metabolic disorders, including periodontitis [3,4,5].

One of these adipokines produced by adipose tissue is leptin. This signaling molecule belongs to the peptide hormones and is mainly produced by white adipose tissue [6]. Through a central feedback mechanism with the mediobasal hypothalamus, it is involved in the control of satiety and, consequently, plays a key role in energy balance [7]. Plasma leptin levels correlate proportionally with total body fat mass [8]. Nevertheless, in obesity, despite high circulating serum leptin levels, a feeling of satiety remains absent. This is described as relative leptin resistance in hypothalamic appetite-associated neurons [9]. Although leptin was originally identified as a regulator of lipid metabolism, its pleiotropic effects are supported by the additional production of leptin in the placenta, bone marrow, skeletal muscle, pituitary gland, and gastric mucosa, as well as the expression of leptin receptors throughout the immune system [10,11].

Increased salivary leptin concentrations were detected in overweight individuals, where also reduced tooth movement was observed compared to normal weight individuals [12].

Orthodontic correction of malocclusions has profound medical relevance, as malocclusions of teeth and jaws not only have aesthetic effects, but are among the most common impairments of oral health and function of the masticatory system in humans. Crowding complicates oral hygiene and is surmised to lead to an increased prevalence of caries [13], gingivitis, and periodontitis [14]. The cause of malocclusions is multifactorial and can be genetic, epigenetic, functional, and environmental [15]. The curative capabilities of orthodontics offer significant potential for establishing and stabilizing oral health by preventing progression of oral disease and dysfunction.

Orthodontic tooth movement is the result of a targeted external mechanical force exerted on the dentoalveolar complex by means of removable or fixed appliances. Orthodontic forces induce compressive and tensile stress in the alveolar bone and periodontal ligament [16,17]. Cells in the periodontal ligament, such as fibroblasts, osteoblasts, and immune cells such as macrophages, are exposed to these forces and are involved in orthodontic force-induced bone remodeling [18,19].

Macrophages are inflammation-mediating immune cells that regulate host defense by phagocytosis of pathogens, cellular debris, and apoptotic cells, and activate the immune response by presenting bacterial antigens and recruiting other inflammatory cells [20]. Macrophages regulate tissue homeostasis in various pathophysiological processes, such as wound healing, hematopoiesis, and malignancy, including the control of metabolic functions. They also play an important role in diseases associated by inflammation-mediated bone resorption [21], such as rheumatoid arthritis and periodontopathies. They also constitute one of the main players in forced tooth movement by interacting with periodontal ligament fibroblasts and promoting the differentiation of osteoclast progenitor cells. Furthermore, macrophages secrete a variety of proinflammatory mediators, such as tumor necrosis factor (TNF), interleukin-1 (IL-1), IL-6, and prostaglandin endoperoxide synthase-2 (PTGS2) in response to mechanical stress [22]. These mediators upregulate the expression of nuclear factor-κB ligand (RANKL) receptor activator, thereby modulating bone resorption [23] and orthodontic tooth movement.

Obesity is a complex disease that, in the face of a growing obese patient population, requires further understanding and awareness to optimize clinical management and ensure the long-term success of orthodontic treatment. Furthermore, leptin increased secretion of proinflammatory mediators in gingival and periodontal ligament fibroblasts without and with mechanical strain [24,25]. Animal studies demonstrated that obesity and leptin reduced orthodontic tooth movement by affecting osteoclastogenesis [26,27]. Leptin exerts regulatory effects on cells of both the innate and adaptive immune systems [28]. However, no studies were available on the effect of leptin on macrophages during mechanical strain. This study improves the understanding of the interplay between mechanical strain and leptin with a focus on immune cells.

Therefore, the focus of this in vitro study was to examine the influence of leptin on the macrophage expression profile of inflammatory mediators during mechanical loading as a model of orthodontic tooth movement.

## 2. Results

### 2.1. Effects of Different Leptin Concentrations without Mechanical Strain

First, we determined effects of different leptin concentrations on cell vitality. We detected no effects of the tested leptin concentrations on cell number. Only after the addition of 10^2^ ng/mL leptin, there was a significant reduction in cell number (*p* = 0.024, Figure 1a). Determination of lactate dehydrogenase (LDH) release showed similar results with no cytotoxic effect of leptin in all tested concentrations (Figure 1b). Analysis of prostaglandin endoperoxide synthase-2 (*Ptgs2*) mRNA expression was significantly increased after the addition of 10^−3^ ng/mL (*p* = 0.021), 10^−1^ ng/mL (*p* < 0.001), and 1 ng/mL leptin (*p* = 0.011; Figure 1c). We decided to use a leptin concentration of 1 ng/mL for further experiments, as we detected no cytotoxic effects and an increased *Ptgs2* gene expression with this concentration. Furthermore, this leptin concentration was determined in gingival fluid [29].

### 2.2. Impact of Leptin on Macrophages in Combination with Porphyromonas Gingivalis LPS

Next, we wanted to test the effect of leptin on the proinflammatory reaction of macrophages without and with additional LPS from *Porphyromonas gingivalis* to mimic inflammation occurring during periodontitis. Gene expression of tumor necrosis factor (*Tnf*) was elevated after treatment with leptin *(p* = 0.028), LPS (*p* = 0.001) and a combination of both (*p* = 0.002; Figure 2a). An addition of LPS to leptin increased the leptin effect (*p* = 0.003), while there was no effect of leptin in comparison with the LPS-only group (*p* = 0.975; Figure 2a). Leptin tended to increase interleukin-1b (*Il1b*) gene expression (*p* = 0.069), while LPS (*p* = 0.003) and the combination of leptin/LPS (*p* < 0.001) showed significant differences (Figure 2b). In the combined group, the upregulation of *Il1b* gene expression was significant compared to leptin (*p* < 0.001) and LPS treatment (*p* = 0.001), indicating a proinflammatory effect of leptin. Accordingly, *Il6* gene expression was increased with leptin (*p* = 0.044) and LPS (*p* < 0.001; Figure 2c). A combination of leptin and LPS increased *Il6* gene expression compared to leptin (*p* < 0.001) or LPS by itself (*p* = 0.006). Gene expression of *Ptgs2* was significantly different to the untreated control after addition of leptin (*p* = 0.012), LPS (*p* < 0.001), and a combination of both (*p* < 0.001; Figure 2d). Again, the combined situation was significantly different to leptin and LPS by itself (*p* < 0.001). Of note, administration of leptin by itself showed proinflammatory effects compared to the untreated control, but these were only slightly pronounced in contrast to the LPS-treated samples.

### 2.3. Impact of Different Leptin Concentrations during Compressive Strain

During orthodontic treatment, mechanical strain is applied on the cells in the periodontal ligament including macrophages. Therefore, we also investigated the impact of leptin during compression. As expected, compression without leptin reduced cell number (*p* < 0.001; Figure 3a). In contrast to cells without compressive strain, leptin reduced cell number significantly at concentrations of 10^3^ ng/mL (*p* = 0.004) and 10^4^ ng/mL (*p* = 0.008). Accordingly, LDH release tended to be upregulated after leptin application at concentrations higher than 10 ng/mL (Figure 3b). We observed increased *Ptgs2* mRNA after compressive strain without additional leptin treatment (*p* < 0.001; Figure 3c). Addition of at least 1 ng/mL leptin (*p* = 0.023) reduced *Ptgs* mRNA compared to compressive strain without leptin (Figure 3c). According to the observed effects on *Ptgs2* mRNA expression and the experiments without mechanical strain, we used a concentration of 1 ng/mL leptin for further experiments with compressive and tensile strain.

### 2.4. Effects of Leptin on the Expression of Proinflammatory Mediators during Compressive Strain

Investigation of tumor necrosis factor (TNF) gene expression (*p* = 0.005, Figure 4a) and protein secretion (*p* = 0.003, Figure 4b) revealed an increase after compression. Addition of leptin did not impact on the observed pressure effect (mRNA: *p* > 0.999; secretion: *p* = 0.485). *Il1b* mRNA was increased after compression (*p* = 0.006; Figure 4c). Treatment with leptin, however, truncated the compressive strain induction of *Il1b* gene expression (*p* = 0.045). Accordingly, we measured increased IL1B secretion after pressure application without leptin (*p* = 0.039) and a reducing effect after addition of leptin (*p* = 0.029; Figure 4d). As expected, IL6 gene expression and secretion was upregulated after compression (*p* < 0.001; Figure 4e,f). Additional treatment with leptin reduced this pressure effect at mRNA and protein levels (*p* = 0.003). For *Ptgs2* mRNA expression, we detected an upregulation after pressure application (*p* < 0.001; Figure 4g). This pressure effect was significantly diminished after addition of leptin (*p* = 0.015). On the protein level, we also detected an increased PTGS2 protein expression in reaction to compressive strain (Figure 4h). After the addition of leptin, this effect was inhibited, indicating anti-inflammatory effects of leptin during pressure application.

### 2.5. Impact of Leptin on the Expression of Proinflammatory Mediators during Tensile Strain

Next to pressure zones in the periodontal ligament, there are zones where the cells in the periodontal ligament are stretched during orthodontic tooth movement. Therefore, we also investigated an effect of leptin on macrophages after tensile strain. Analysis of TNF mRNA expression (Figure 5a) and protein secretion (Figure 5b) revealed an increase due to tensile strain (mRNA: *p* < 0.001; secretion: *p* = 0.003). Contrary to the compressive strain, this effect of stretching was diminished with additional leptin treatment (mRNA: *p* = 0.023; secretion: *p* = 0.038). We detected no upregulation of IL1B mRNA expression (*p* = 0.627; Figure 5c) or protein secretion (*p* = 0.925; Figure 5d) after tensile force application. Comparable to compressive strain, tension increased IL6 mRNA levels and protein secretion (*p* < 0.001; Figure 5e,f). This effect was abolished by additional leptin treatment (*p* < 0.001). *Ptgs2* mRNA levels were elevated with tensile strain (*p* = 0.025) and reduced to the untreated level with additional leptin treatment (*p* = 0.007; Figure 5g). This effect was also observed at the protein level (Figure 5h).

## 3. Discussion

Obesity is a growing problem in adults and children; thereby, orthodontics is also confronted with overweight patients [2]. Obesity is one of the most important risk factors for inflammatory processes in the periodontal ligament [5], due to elevated secretion of adipokines such as leptin [30,31]. It is already known that leptin can enhance the release of proinflammatory factors in periodontal and gingival fibroblasts under control conditions and exposure to mechanical strain [24,25]. Next to fibroblasts, immune cells in the periodontal ligament, such as macrophages, are exposed to mechanical stress as well and can, therefore, modulate the immune response. Therefore, the purpose of this study was to investigate the effect of leptin without and with mechanical strain on macrophages.

Without additional mechanical loading, leptin alone only showed a slight cytotoxic effect on RAW264.7 macrophages at a concentration of 10^2^ ng/mL. Leptin levels in gingival crevicular fluid range from 1.1 to 2.3 ng/mL [29] in dependence of periodontal health status. With this concentration, no cytotoxic effect was obvious; however, we detected an enhancement in the expression of the proinflammatory gene *Ptgs2*. Therefore, we decided to use a concentration of 1 ng/mL leptin for the following experiments.

With this concentration, we detected an increased expression of inflammatory genes in RAW264.7 macrophages without additional mechanical strain. This was in line with the literature, as leptin was reported to impact on the function of macrophages by modulating the microbicidal response and macrophage polarization [32,33]. Tsiortra et al. reported increased secretion of TNF, IL6, and IL1B in human mononuclear cells after treatment with leptin in vitro, which was in line with our data [34]. In general, leptin signaling appears to be required in the maintenance of phagocytic functions and impacts on the production of inflammatory mediators by macrophages [32].

Stimulation with *Porphorymonas gingivalis* (*P. gingivalis*) lipopolysaccharides (LPS) increased the expression of inflammatory genes by RAW264.7 macrophages, which was in line with the results presented by Park et al. [35]. Most studies dealing with inflammation were performed with *Escherichia coli* LPS; however, as we were interested in processes occurring in the oral cavity, we decided to use *P. gingivalis* LPS, as this Gram-negative bacterium is a major periodontopathic pathogen [36]. Diya et al. demonstrated that, in contrast to *E. coli.* LPS, *P. gingivalis* LPS promote the production of cytokines by induction of the TLR2–JNK signaling pathway [37]. Treatment with leptin additionally elevated the increasing effects of *P. gingivalis* LPS on inflammatory mediators. Leptin was reported to upregulate [38] and downregulate [39,40] LPS-induced effects.

During orthodontic tooth movement pressure and tension zones develop in the periodontal ligament [16,17]. This mechanical stress affects the secretion of inflammatory mediators by periodontal ligament fibroblasts [18] and macrophages, as already reported previously [41]. According to untreated cells without compressive strain, we only detected slight cytotoxic effects with a leptin concentration of 10^2^ ng/mL. The physiological leptin concentration of 1 ng/mL, however, reduced the induction of Ptgs2 gene expression evoked by compressive strain. As mentioned before, additional leptin was reported to have upregulating and downregulating effects, depending on the cell type, concentration, and effect investigated. An in vitro study in periodontal ligament fibroblasts (PDLF) showed no changes in gene expression or secretion of inflammatory mediators with the physiological leptin concentration of 1 ng/mL used in this study. Only an excessive concentration of 10^4^ ng/mL leptin increased the expression and secretion of inflammatory mediators under control and pressure conditions [24]. In this study, we clearly see an inhibitory effect on the expression of inflammatory factors in macrophages, which might result in a reduced extent of sterile inflammation, and thereby, orthodontic tooth movement. This was in line with animal studies, which showed that both obesity and leptin reduced orthodontic tooth movement in mice and rats by affecting osteoclastogenesis [26,27].

Leptin can impact on the expression profile of macrophages under control and pressure conditions in different ways, as it can promote a pro-inflammatory effect without mechanical stress or an anti-inflammatory phenotype after mechanical stress. The underlying molecular mechanisms must be subject of further research to better understand the mode of action of leptin and its mechanical force on macrophages.

## 4. Materials and Methods

### 4.1. Cell Culture Experiments

Experiment 1: For the determination of the optimal leptin concentration for further experiments, 10^6^ RAW264.7 macrophages (Cell Lines Service, Eppelheim, Germany) per well were seeded on a polystyrene 6-well plate (83.3920, Sarstedt, Nürnbrecht, Germany) in RPMI Medium 1640 with GlutaMAX-I (61870-010, Thermo Fisher Scientific, Waltham, MA, USA), supplemented with 10% fetal bovine serum (P30-3302, PAN-Biotech, Aidenbach, Germany) and 1% antibioticum/antimycoticum (A5955, Sigma-Aldrich, St. Louis, MO, USA). Different amounts of recombinant mouse leptin (cyt-351, Prospec, East Brunswick, NJ, USA) were added ranging from 10^−4^ to 10^4^ ng/mL. After 24 h, cells were either left untreated of exposed to compressive strain by application of a sterile glass plate (2 g/cm^2^) for a further 4 h [18,22]. After that cell number, cytotoxicity and gene expression were examined.

Experiment 2: To assess the effects of leptin under control conditions and parallel stimulation with *Porphyromonas gingivalis* (*P. gingivalis*) LPS, 10^6^ RAW264.7 macrophages per well were seeded on a polystyrene 6-well plate (83.3920, Sarstedt, Nürnbrecht, Germany) in RPMI Medium 1640 with GlutaMAX-I (61870-010, Thermo Fisher Scientific, Waltham, MA, USA) supplemented with 10% fetal bovine serum (P30-3302, PAN-Biotech, Aidenbach, Germany) and 1% antibioticum/antimycoticum (A5955, Sigma-Aldrich, St. Louis, MO, USA) and either treated with leptin (1 ng/mL; cyt-351, Prospec, East Brunswick, NJ, USA), *P. gingivalis* LPS (5 µg/mL; tlrl-pglps, InvivoGen, San Diego, CA, USA), a combination of both, or left untreated as control. After incubation overnight, the cell number (Appendix A), cytotoxicity (Appendix A), and gene expression were examined.

Experiment 3: To investigate effects of leptin on compressive strain 10^6^ RAW264.7 macrophages per well were seeded on a polystyrene 6-well plate (83.3920, Sarstedt, Nürnbrecht, Germany) in RPMI Medium 1640 with GlutaMAX-I (61870-010, Thermo Fisher Scientific, Waltham, MA, USA) supplemented with 10% fetal bovine serum (P30-3302, PAN-Biotech, Aidenbach, Germany) and 1% antibioticum/antimycoticum (A5955, Sigma-Aldrich, St. Louis, MO, USA), and both were treated with leptin (1 ng/mL; cyt-351, Prospec, East Brunswick, NJ, USA) or left untreated for 24 h. After that, macrophages were exposed to compressive strain by application of a sterile glass plate (2 g/cm^2^) for a further 4 h or left untreated [18,22]. After compressive strain cell number (Appendix A), cytotoxicity (Appendix A), and gene as well as protein expression were determined.

Experiment 4: To investigate effects of leptin on tensile strain 10^6^ RAW264.7 macrophages per well of a 6-well bioflex plate (BF-3001U, Dunn Labortechnik, Asbach, Germany) were seeded in RPMI Medium 1640 with GlutaMAX-I (61870-010, Thermo Fisher Scientific, Waltham, MA, USA) supplemented with 10% fetal bovine serum (P30-3302, PAN-Biotech, Aidenbach, Germany) and 1% antibioticum/antimycoticum (A5955, Sigma-Aldrich, St. Louis, MO, USA), and both were treated with leptin (1 ng/mL; cyt-351, Prospec, East Brunswick, NJ, USA) or left untreated for 24 h. After that, macrophages were exposed to tensile strain by insertion of a spherical silicone stamp (16%) for a further 4 h or left untreated [22,42]. After tensile strain, the cell number (Appendix A), cytotoxicity (Appendix A), and gene as well as protein expression were determined.

### 4.2. Determination of Cell Number

After the appropriate incubation times, cell culture plates were placed on ice and the supernatant of cell culture medium was removed. Adhered cells were scraped off the plate in 1 mL PBS using a cell scraper. From this cell suspension, 100 µL were pipetted into 10 mL of 0.8% NaCl and mixed. The cell number was assessed in the Coulter Counter (Z2, Beckham Coulter, Brea, CA, USA) with a size of 8–15 µm.

### 4.3. Determination of Cytotoxicity by Lactate Dehydrogenase (LDH) Assay

LDH assay (4744926001, Roche, Penzberg, Germany) was performed according to the manufacturer’s instructions and evaluated in the ELISA Reader (Multiscan GO, Thermo Fisher Scientific, Waltham, MA, USA).

### 4.4. RNA Isolation and cDNA Synthesis

For RNA isolation cells were scraped off in 1 mL PBS and centrifuged at 1500 rpm for 5 min at 4 °C. The cell pellet was resuspended in 500 µL RNA solv (R6830-01, VWR international, Radnor, PA, USA). Samples were mixed with 100 µL of chloroform and vortexed for 30 sec. After incubation on ice for 15 min, the suspension was centrifuged at 13,000 rpm for 15 min at 4 °C. The aqueous phase was transferred to a new tube containing 500 µL isopropanol (20.842.330, VWR international, Radnor, PA, USA). After inverting, samples incubated at −80 °C overnight. This was followed by centrifugation at 13,000 rpm for 30 min at 4 °C and washing with 80% ethanol. The resulting RNA pellet was dried for 30 min and resuspended in 20 µL RNase-free H_2_O_dd_ (T143.5, Carl Roth, Karlsruhe, Germany). For quantitative analysis, the measured RNA (Nanophotometer N60, Implen, Munich, Gemrany) was diluted to 100 ng/mL with RNase-free H_2_O_dd_ (T143.5, Carl Roth, Karlsruhe, Germany). The reaction was performed in a total volume of 10 µL, containing 5.5 µL of diluted RNA and 4.5 µL of master mix consisting of 2 µL M-MLV buffer (M531A, Promega, Madison, WI, USA), 0.5 µL oligo_dT_ (SO123, Thermo Fisher Scientific, Waltham, MA, USA), 0.5 µL random hexamer (SO142, Thermo Fisher Scientific, Waltham, MA, USA), 0.5 µL RNase inhibitor (EO0381, Thermo Fisher Scientific, Waltham, MA, USA), and 0.5 µL M-MLV reverse transcriptase (M170B, Promega, Madison, WI, USA). The thermocycler (Thermocycler Tone 96 G-Biometra, Analytik Jena, Jena, Germany) program for reverse transcription consisted of reverse transcription for one hour at 37 °C and denaturation of reverse transcriptase, and thus, termination of the RT reaction at 95 °C for 2 min. Subsequently, the synthesized cDNA was diluted 1:10 with RNase-free H_2_O_dd_ (T143.5, Carl Roth, Karlsruhe, Germany).

### 4.5. RT-qPCR Analysis

Real-time polymerase chain reaction (RT-qPCR) was used to analyze mRNA expression profile after exposure to leptin under additional compressive or tensile strain. We used a combination of *Eef1a1/Sdha* for pressure experiments and *Gapdh/Tbp* for the stretching experiments [22]. All qPCRs were performed in duplicate. For this, 1.5 µL cDNA was pipetted into a 96-well plate (712282, Biozym, Hessisch Oldendorf, Germany), followed by 8.5 µL of primer mix consisting of 0.25 µL forward primer, 0.25 µL reverse primer (Table 1), 3 µL RNase-free H_2_O_dd_ (T143.5, Carl Roth, Karlsruhe, Germany), and 5 µL Luna Universal qPCR Master Mix (M3003E, New England Biolabs, Ipswich, MA, USA) for each sample. The 96-well plates were covered with optical film (T12350, Biozym, Hessisch Oldendorf, Germany). The plates were read in the Realplex^2^ (Eppendorf, Hamburg, Germany) using the following program (95 °C for 2 min, 45 cycles each of 95 °C for 10 s, 60 °C for 20 s, and 72 °C for 8 s). To determine the relative gene expression, the formula 2^−ΔCq^ was used [43].

### 4.6. Western Blot Analysis

After discarding the cell culture supernatant, 100 µL CellLytic (C2978, Sigma-Aldrich, St. Louis, MO, USA) supplemented with proteinase inhibitor (87786, Thermo Fisher Scientific, Waltham, MA, USA) was applied to each well. Cells were detached from the bottom using a cell scraper and incubated on ice for 15 min. After centrifugation at 13,000 rpm for 15 min at 4 °C the supernatant was transferred to a new tube and protein concentration was determined with Roti-Quant (K015.3, Carl Roth, Karlsruhe, Germany), according to the manufacturer’s instructions. A defined amount of protein was mixed with sample buffer (3.75 mL TRis/HCl pH 6.8 (Sigma-Aldrich, St. Louis, MO, USA), 3 mL glycerol (3783.1, Carl Roth, Karlsruhe, Germany), 1.2 g SDS (8029.1, Carl Roth, Karlsruhe, Germany), 0.06 g bromophenol blue (B-5525, Sigma-Aldrich, St. Louis, MO, USA), and 0.15 g DTT (A2948.0005, AppliChem, Darmstadt, Germany) in a final volume of 10 mL) and heated to 70 °C for 7 min. This was followed by a 7 min incubation on ice to stop protein denaturation. After centrifugation (7000 rpm, 7 min, 4 °C) the samples were stored at −80 °C until use. Proteins were separated on 12% polyacrylamid gels and transferred to polyvinylidene difluorid membranes (T830.1, Carl Roth, Karlsruhe, Germany). Membranes were blocked with 5% milk (T145.3, Carl Roth, Karlsruhe, Germany) in TBS-T for 1 h at room temperature. The membranes were incubated in specific primary antibody, i.e., PTGS2 (PA5-16817, Thermo Fisher Scientific, Waltham, MA, USA) and ACTIN (A2066, Sigma-Aldrich, St. Louis, MO, USA), overnight at 4 °C on a roller mixer. The membranes were washed three times with TBS-T for 10 min each and incubated with secondary antibody (611-1302, Rockland Immunochemicals, Gilbertsville, PA, USA) for 1 h at room temperature. After three further washes for 10 min each in TBS-T, the final step was the detection of the proteins immobilized on the PVDF membrane by a chemiluminescence reaction catalyzed by the secondary antibody. In preparation for this, the membrane was wetted with Immobilon Forte (WBLUF0100, Sigma-Aldrich, St. Louis, MO, USA), as a substrate for HRP, and incubated for 3 min. Documentation was performed using the VWR Genoplex system (VWR international, Radnor, PA, USA).

### 4.7. Enzyme-Linked Immunosorbent Assays (ELISA)

Cell culture supernatants were stored at −80 °C until use. The supernatants were carefully thawed on ice and centrifuged. IL6 (MBS335514, MyBiosource, San Diego, CA, USA), IL1B (MBS412296, MyBiosource, San Diego, CA, USA), and TNF (MBS335449, My Biosource, San Diego, CA, USA). The commercially available ELISAs contained all necessary reagents und were performed according to the manufacturer’s instructions. Data were normalized to the control group without leptin to obtain relative differences.

### 4.8. Statistics

Statistical analysis was performed using GraphPad Prism version 9.2 (GraphPad software, San Diego, CA, USA). To analyze the datasets, the normal distribution of the data was examined with the Shapiro–Wilk test. A Brown–Forsythe test was performed to determine homogeneity of variance. Subsequently, Welch-corrected ANOVA followed either by T-tests for determination of leptin concentration or Dunnett’s T3 post hoc test were performed. If the *p*-value was less than 0.05, the differences were considered statistically significant. In the graphs, mean and standard deviation are presented.

## 5. Conclusions

Leptin has an influence on the expression profile of macrophages that is strongly dependent on the mechanical load of the cells. The mechanism of this differential leptin effect should be further investigated.

## Figures and Tables

**Figure 1 ijms-23-10727-f001:**
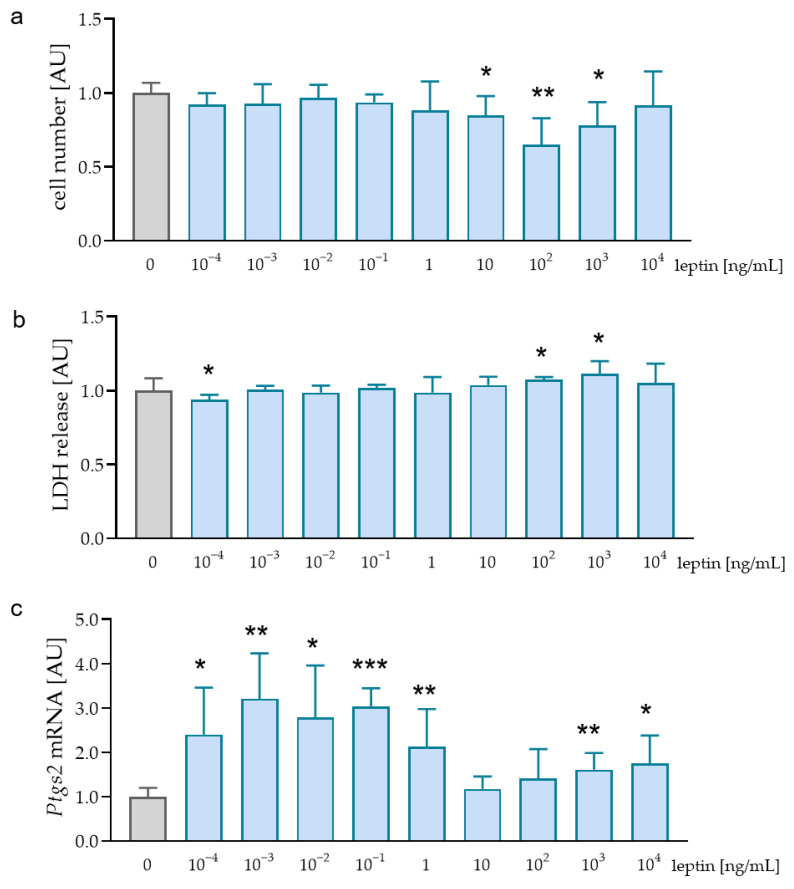
Cell number (**a**), lactate dehydrogenase (LDH) release (**b**), and *Ptgs2* gene expression (**c**) after treatment of RAW264.7 macrophages with different leptin concentrations. n ≥ 5; Statistics: ANOVA followed by unpaired t-tests with Welch-correction. * *p* < 0.05; ** *p* < 0.01; *** *p* < 0.001 compared to 0 ng/mL leptin.

**Figure 2 ijms-23-10727-f002:**
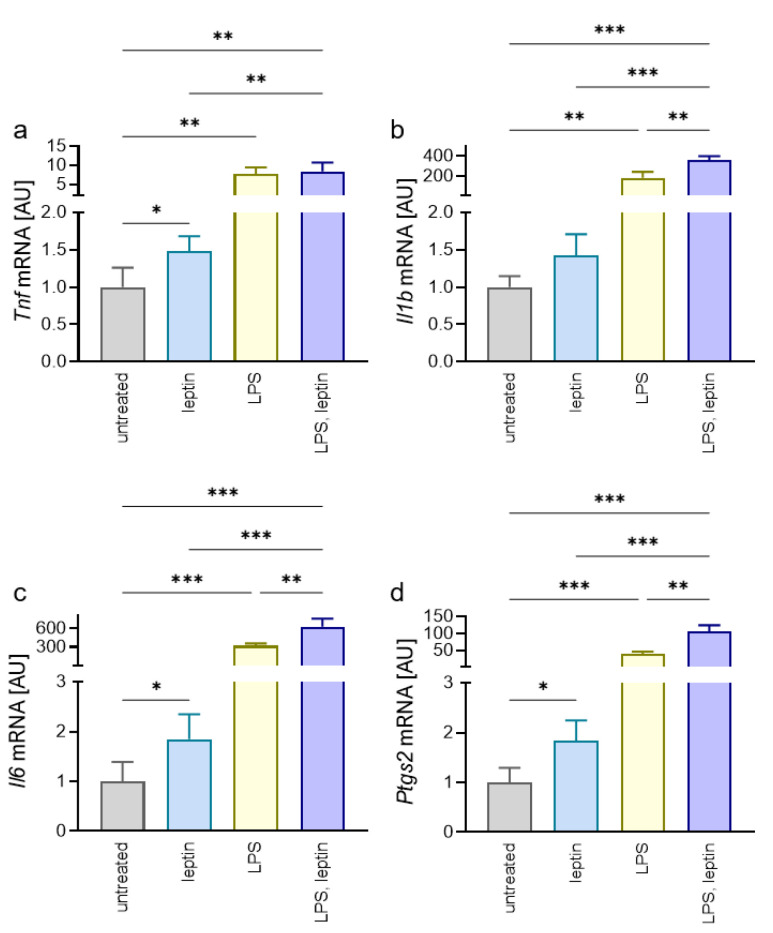
Impact of leptin, *Porphyromonas gingivalis* LPS and a combination of both on gene expression of *Tnf* (**a**), *Il1b* (**b**), *Il6* (**c**), and *Ptgs2* (**d**) in RAW264.7 macrophages; n = 6; Statistics: Welch-corrected ANOVA with Dunnett’s T3 multiple comparisons test; * *p* < 0.05, ** *p* < 0.01, *** *p* < 0.001.

**Figure 3 ijms-23-10727-f003:**
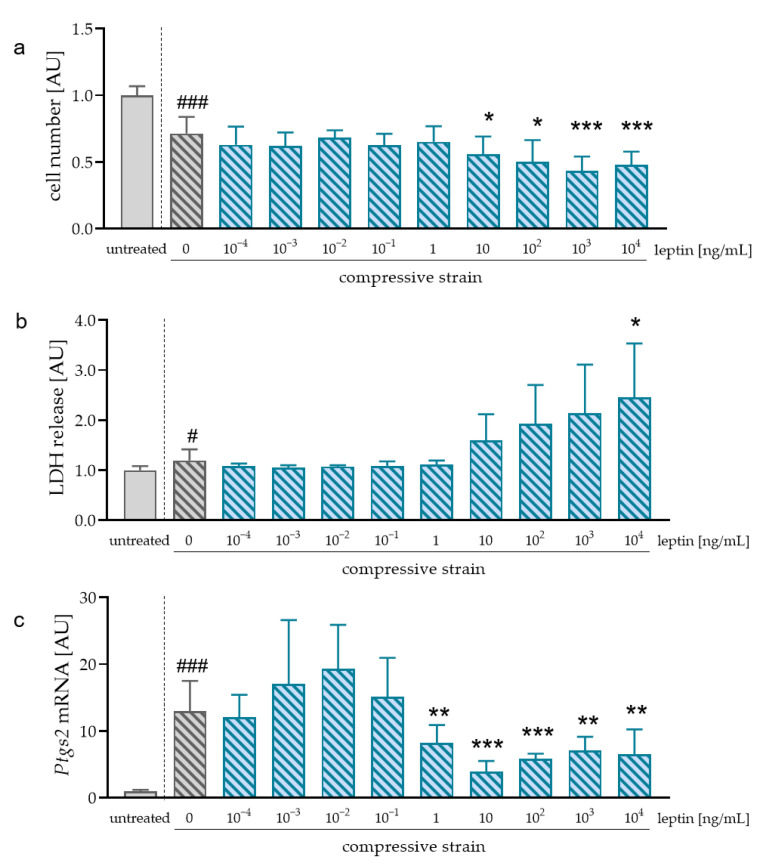
Cell number (**a**), LDH release (**b**) and *Ptgs2* gene expression (**c**) after treatment of RAW264.7 macrophages with different leptin concentrations. n > 5; Statistics: ANOVA followed by unpaired t tests with Welch-correction. ^#^
*p* < 0.05; ^###^
*p* < 0.001 compared to untreated control; * *p* < 0.05; ** *p* < 0.01; *** *p* < 0.001 compared to 0 ng/mL leptin and compressive strain.

**Figure 4 ijms-23-10727-f004:**
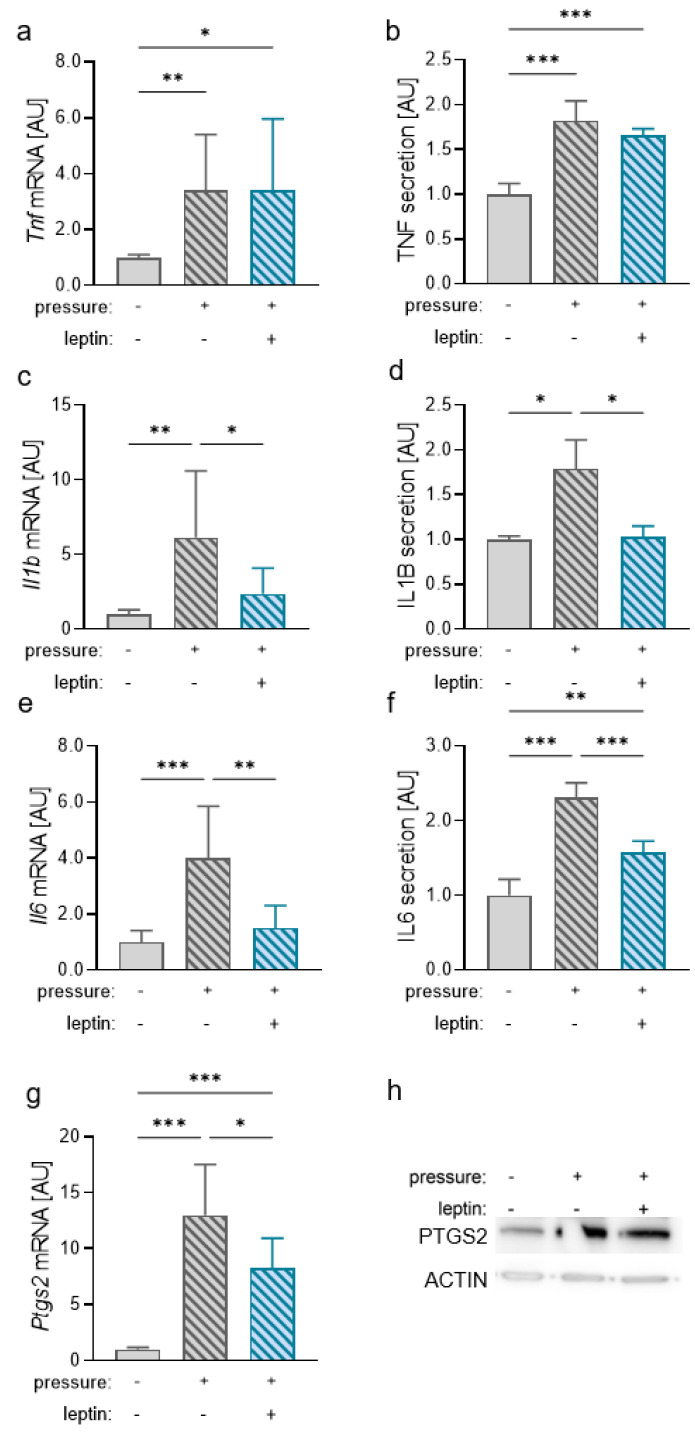
Impact of leptin in combination with compressive strain on TNF (**a**,**b**), IL1B (**c**,**d**), IL6 (**e**,**f**), and PTGS2 (**g**,**h**) gene and protein expression and secretion in RAW264.7 macrophages; n ≥ 4; Welch-corrected ANOVA with Dunnett’s T3 multiple comparisons test. * *p* < 0.05; ** *p* < 0.01; *** *p* < 0.001. Uncropped blot is presented in Appendix A.

**Figure 5 ijms-23-10727-f005:**
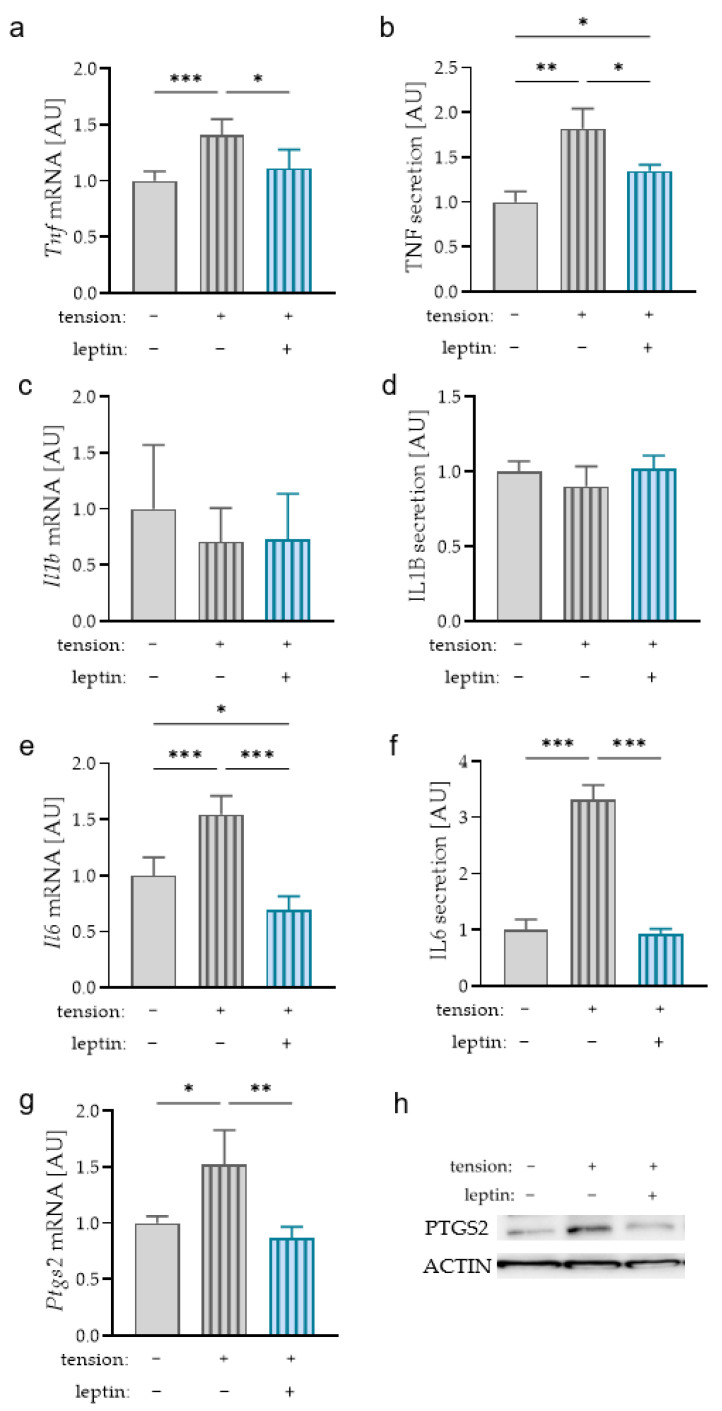
Impact of leptin in combination with tensile strain on TNF (**a**,**b**), IL1B (**c**,**d**), IL6 (**e**,**f**), and PTGS2 (**g**,**h**) gene and protein expression and secretion in RAW264.7 macrophages; n ≥ 4; Welch-corrected ANOVA with Dunnett’s T3 multiple comparisons test. * *p* < 0.05; ** *p* < 0.01; *** *p* < 0.001. Uncropped blot is presented in Appendix A.

**Table 1 ijms-23-10727-t001:** Primer sequences for reference genes (pressure: *Eef1a1/Sdha*; tension: *Gapdh/Tbp*) and target genes.

Symbol	Gene Name	5′-Forward Primer-3′	5′-Reverse Primer-3′
*Eef1a1*	Eukaryotic Translation Elongation Factor-1-α-1	AAAACATGATTACAGGCACATCCC	GCCCGTTCTTGGAGATACCAG
*Gapdh*	Glyceraldehyde-3-phosphate dehydrogenase	GTCATCCCAGAGCTGAACGG	ATGCCTGCTTCACCACCTTC
*Il1b*	Interleukin-1-beta	GTGTAATGAAAGACGGCACACC	ACCAGTTGGGGAACTCTGC
*Il6*	Interleukin-6	ACAAAGCCAGAGTCCTTCAGAG	GAGCATTGGAAATTGGGGTAGG
*Ptgs2*	prostaglandin-endoperoxide synthase-2	TCCCTGAAGCCGTACACATC	TCCCCAAAGATAGCATCTGGAC
*Sdha*	Succinate Dehydrogenase Complex Flavoprotein Subunit A	AACACTGGAGGAAGCACACC	AGTAGGAGCGGATAGCAGGAG
*Tbp*	TATA-box-binding protein	CTATCACTCCTGCCACACCAG	CACGAAGTGCAATGGTCTTTAGG
*Tnf*	Tumor necrosis factor	ACAAGCCTGTAGCCCACGTC	TTGTTGTCTTTGAGATCCATGCC

## Data Availability

All datasets are publically available either as Appendix A to this article or upon request from the corresponding author.

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
