# Peer review of "Impact of Leptin on the Expression Profile of Macrophages during Mechanical Strain In Vitro"

_ijms, 2022, doi:10.3390/ijms231810727_

Round 1

Reviewer 1 Report

Dear authors and editors, it was a pleasure to revise this manuscript. However, some minor issues have to be highlighted: 

1. Title: add the study design

2. Abstract: Add the study design and clarify the aim

3. M&M: Did you apply the Bonferroni correction when testing multiple variables? If yes, please specify. 

Best regards. 

Author Response

Comments and Suggestions for Authors

Dear authors and editors, it was a pleasure to revise this manuscript. However, some minor issues have to be highlighted: 

  1. Title: add the study design

We indicated the study design in the title.

Revised title: Impact of leptin on the expression profile of macrophages during mechanical strain in vitro

  1. Abstract: Add the study design and clarify the aim

As suggested, we added the study design and the aim of the study.

Revised test: Therefore, the aim of this in vitro study was to assess the influence of leptin on the expression profile of macrophages during simulated orthodontic treatment.

  1. M&M: Did you apply the Bonferroni correction when testing multiple variables? If yes, please specify.

We didn´t perform a Bonferroni correction, but a Dunnett´s T3 multi comparison test.

Reviewer 2 Report

In the manuscript: “Impact of leptin on macrophages during mechanical strain”, the authors investigated the influence of leptin on macrophage expression profile of inflammatory mediators during mechanical loading as a model of orthodontic tooth movement. The article is interesting and had promising results regarding one of the consequences of obesity, a complex disease, with an alarming increase in the number of patients, but I have some remarks that I think the authors must be considered before publication:

1.     The abstract must be re-organized, because, in the introduction the authors refer also to obesity. So, this disease, must be also mentioned in the abstract. It's true, that they have written at line 19: “Leptin levels are elevated in obesity and chronic inflammatory responses”, but this is not enough. The link between obesity, leptin and Orthodontic problems must be highlighted. 

2.     Also, the title is too general, from my point of view.

3.     The whole introduction must be re-organized. The presented information is fine, but the paragraphs are not clearly connected. E.g., the last two paragraphs (those referring to obesity) must be move at the beginning of the introduction, as leptin, the studied adipokine, is produce by adipose tissue, so is strongly related with obesity. 

4.     It is very important to mention in the introduction, the novelty of the study. Given the presumption that obesity and leptin reduced orthodontic tooth movement, by affecting osteoclastogenesis, has already been validated in some in vivo studies, why an in vitro study is still necessary? What does it bring new to the field?

5.     Even if Conclusion section is not mandatory, I recommend you to add it, in order to properly summarize your main results. Also, what are your perspectives?

Author Response

Comments and Suggestions for Authors

In the manuscript: “Impact of leptin on macrophages during mechanical strain”, the authors investigated the influence of leptin on macrophage expression profile of inflammatory mediators during mechanical loading as a model of orthodontic tooth movement. The article is interesting and had promising results regarding one of the consequences of obesity, a complex disease, with an alarming increase in the number of patients, but I have some remarks that I think the authors must be considered before publication:

According to reviewers suggestion, we checked the manuscript for English style.

  1. The abstract must be re-organized, because, in the introduction the authors refer also to obesity. So, this disease, must be also mentioned in the abstract. It's true, that they have written at line 19:“Leptin levels are elevated in obesity and chronic inflammatory responses”, but this is not enough. The link between obesity, leptin and Orthodontic problems must be highlighted.

According to the reviewer’s suggestion, we adjusted the abstract.

Revised text: Childhood obesity is a growing problem in industrial societies and associated with increased leptin levels in serum and salvia. Orthodontic treatment provokes pressure and tension zones within the periodontal ligament, where, in addition to fibroblasts, macrophages are exposed to these mechanical loadings. Given the increasing number of orthodontic patients with these conditions, insights into the effects of elevated leptin levels on the expression profile of macrophages during mechanical strain are of clinical interest. Therefore, the aim of this in vitro study was to assess the influence of leptin on the expression profile of macrophages during simulated orthodontic treatment.

  1. Also, the title is too general, from my point of view.

According to the reviewer’s suggestion, we adjusted the title.

Revised title: Impact of leptin on the expression profile of macrophages during mechanical strain in vitro.

  1. The whole introduction must be re-organized. The presented information is fine, but the paragraphs are not clearly connected. E.g., the last two paragraphs (those referring to obesity) must be move at the beginning of the introduction, as leptin, the studied adipokine, is produce by adipose tissue, so is strongly related with obesity.

According to the reviewer’s suggestion, we reorganized the introduction and focussed on the role of leptin during orthodontic treatment.

  1. It is very important to mention in the introduction, the novelty of the study. Given the presumption that obesity and leptin reduced orthodontic tooth movement, by affecting osteoclastogenesis, has already been validated in some in vivo studies, why an in vitro study is still necessary? What does it bring new to the field?

According to the reviewer’s suggestion, we adjusted the title.

Revised text: Obesity is a complex disease that, in the face of a growing obese patient population, requires further understanding and awareness to optimize clinical management and ensure the long-term success of orthodontic treatment. Furthermore, leptin increased secretion of proinflammatory mediators in gingival and periodontal ligament fibroblasts without and with mechanical strain [24,25]. Animal studies demonstrated that obesity and leptin reduced orthodontic tooth movement by affecting osteoclastogenesis [26,27]. Leptin exerts regulatory effects on cells of both the innate and adaptive immune systems [28]. However, no studies were available focussing on the effect of leptin on macrophages during mechanical strain. This study advances the understanding of the interplay between mechanical strain and leptin with a focus on immune cells.

  1. Even if Conclusion section is not mandatory, I recommend you to add it, in order to properly summarize your main results. Also, what are your perspectives?

As suggested, we added a conclusion to the manuscript.

Revised text: Leptin has an influence on the expression profile of macrophages that is strongly dependent on the mechanical load of the cells. The mechanism of this differential leptin effect should be further investigated.

Comments and Suggestions for Authors

In the manuscript: “Impact of leptin on macrophages during mechanical strain”, the authors investigated the influence of leptin on macrophage expression profile of inflammatory mediators during mechanical loading as a model of orthodontic tooth movement. The article is interesting and had promising results regarding one of the consequences of obesity, a complex disease, with an alarming increase in the number of patients, but I have some remarks that I think the authors must be considered before publication:

According to reviewers suggestion we checked the manuscript for English style.

  1. The abstract must be re-organized, because, in the introduction the authors refer also to obesity. So, this disease, must be also mentioned in the abstract. It's true, that they have written at line 19:“Leptin levels are elevated in obesity and chronic inflammatory responses”, but this is not enough. The link between obesity, leptin and Orthodontic problems must be highlighted.

According to the reviewer’s suggestion, we adjusted the abstract.

Revised text: Childhood obesity is a growing problem in industrial societies and associated with increased leptin levels in serum and salvia. Orthodontic treatment provokes pressure and tension zones within the periodontal ligament, where, in addition to fibroblasts, macrophages are exposed to these mechanical loadings. Given the increasing number of orthodontic patients with these conditions, insights into the effects of elevated leptin levels on the expression profile of macrophages during mechanical strain are of clinical interest. Therefore, the aim of this in vitro study was to assess the influence of leptin on the expression profile of macrophages during simulated orthodontic treatment.

  1. Also, the title is too general, from my point of view.

According to the reviewer’s suggestion, we adjusted the title.

Revised title: Impact of leptin on the expression profile of macrophages during mechanical strain in vitro.

  1. The whole introduction must be re-organized. The presented information is fine, but the paragraphs are not clearly connected. E.g., the last two paragraphs (those referring to obesity) must be move at the beginning of the introduction, as leptin, the studied adipokine, is produce by adipose tissue, so is strongly related with obesity.

According to the reviewer’s suggestion, we reorganized the introduction and focussed on the role of leptin during orthodontic treatment.

  1. It is very important to mention in the introduction, the novelty of the study. Given the presumption that obesity and leptin reduced orthodontic tooth movement, by affecting osteoclastogenesis, has already been validated in some in vivo studies, why an in vitro study is still necessary? What does it bring new to the field?

According to the reviewer’s suggestion, we adjusted the title.

Revised text: Obesity is a complex disease that, in the face of a growing obese patient population, requires further understanding and awareness to optimize clinical management and ensure the long-term success of orthodontic treatment. Furthermore, leptin increased secretion of proinflammatory mediators in gingival and periodontal ligament fibroblasts without and with mechanical strain [24,25]. Animal studies demonstrated that obesity and leptin reduced orthodontic tooth movement by affecting osteoclastogenesis [26,27]. Leptin exerts regulatory effects on cells of both the innate and adaptive immune systems [28]. However, no studies were available focussing on the effect of leptin on macrophages during mechanical strain. This study advances the understanding of the interplay between mechanical strain and leptin with a focus on immune cells.

  1. Even if Conclusion section is not mandatory, I recommend you to add it, in order to properly summarize your main results. Also, what are your perspectives?

As suggested, we added a conclusion to the manuscript.

Revised text: Leptin has an influence on the expression profile of macrophages that is strongly dependent on the mechanical load of the cells. The mechanism of this differential leptin effect should be further investigated.

Reviewer 3 Report

The macrophages and adipocytes in adipose tissue are major proinflammatory cytokines sources in obese individuals. Thus, obesity is associated with chronic pro-inflammatory signaling pathways and why obesity increases an individual’s risk of developing inflammatory-based diseases, including gingival and periodontal ligament diseases. The work: “Impact of leptin on macrophages during mechanical strain” is a part of the above trend of scientific research. The results of leptin's influence on macrophages in combination with Porphyromonas gingivalis LPS - were predictable.

To make the reviewed work more valuable it needs to be detailed:

1.      Try to explain why leptin at 10 4 ng/ml has no effect on cell number and lactatdehydrogenase (LDH) release.

2.      Try to explain how obesity (associated with an increase in leptin) affects orthodontic tooth movement disorders.

3.      Describe  in more detail the Elisa tests, that were used.

4.      Numbering of cited references is mixed up (30,31 - before 27,28,29)

Author Response

Comments and Suggestions for Authors

The macrophages and adipocytes in adipose tissue are major proinflammatory cytokines sources in obese individuals. Thus, obesity is associated with chronic pro-inflammatory signalling pathways and why obesity increases an individual’s risk of developing inflammatory-based diseases, including gingival and periodontal ligament diseases. The work: “Impact of leptin on macrophages during mechanical strain” is a part of the above trend of scientific research. The results of leptin's influence on macrophages in combination with Porphyromonas gingivalis LPS - were predictable.

To make the reviewed work more valuable it needs to be detailed:

  1. Try to explain why leptin at 104 ng/ml has no effect on cell number and lactatdehydrogenase (LDH) release.

There are only slight but significant changes in cell number and LDH release with 102 and 103 ng/ml leptin. With 104 ng/ml leptin without mechanical strain the differences were no longer significant. But the LDH levels were still slightly increased to 1.054 compared to the untreated control. For the experiments with mechanical strain, we used lower leptin concentrations to ensure no cytotoxic effects.

  1. Try to explain how obesity (associated with an increase in leptin) affects orthodontic tooth movement disorders.

As suggested, we adjusted the introduction and focus on the association of leptin and orthodontics.

Revised text: Obesity is a complex disease that, in the face of a growing obese patient population, requires further understanding and awareness to optimize clinical management and ensure the long-term success of orthodontic treatment. Furthermore, leptin increased secretion of proinflammatory mediators in gingival and periodontal ligament fibroblasts without and with mechanical strain [24,25]. Animal studies demonstrated that obesity and leptin reduced orthodontic tooth movement by affecting osteoclastogenesis [26,27]. Leptin exerts regulatory effects on cells of both the innate and adaptive immune systems [28]. However, no studies were available focussing on the effect of leptin on macrophages during mechanical strain. This study improves the understanding of the interplay between mechanical strain and leptin with a focus on immune cells.

Therefore, the focus of this in vitro study was to examine the influence of leptin on macrophage expression profile of inflammatory mediators during mechanical loading as a model of orthodontic tooth movement.

  1. Describe in more detail the Elisa tests, that were used.

Cell culture supernatants were stored at -80°C until use. The supernatants were carefully thawed on ice and centrifuged. IL6 (MBS335514, MyBiosource, San Diego, CA,USA), IL1B (MBS412296, MyBiosource, San Diego, CA,USA) and TNF (MBS335449, MyBiosource, San Diego, CA,USA) The commercially available ELISAs contained all necessary reagents und were performed according to the manufacturer's instructions. Data were normalized to the control group without leptin to obtain relative differences.

  1. Numbering of cited references is mixed up (30,31 - before 27,28,29)

Thank you for this point, we again checked and actualised the numbering von the cited references.

Round 2

Reviewer 2 Report

The authors addressed all my comments. The manuscript looks better now and can be published in the present form.